# Landslide Inventory (2001–2017) of Chittagong Hilly Areas, Bangladesh

**Yasin Wahid Rabby *** and **Yingkui Li**

Department of Geography, University of Tennessee, Knoxville, TN 37916, USA; yli32@utk.edu
* Correspondence: yrabby@vols.utk.edu; Tel.: +1-865-455-0269

**Abstract:** Landslides are a frequent natural hazard in Chittagong Hilly Areas (CHA), Bangladesh, which causes the loss of lives and damage to the economy. Despite this, an official landslide inventory is still lacking in this area. In this paper, we present a landslide inventory of this area prepared using the visual interpretation of Google Earth images (Google Earth Mapping), field mapping, and a literature search. We mapped 730 landslides that occurred from January 2001 to March 2017. Different landslide attributes including type, size, distribution, state, water content, and triggers are presented in the dataset. In this area, slide and flow were the two dominant types of landslides. Out of the five districts (Bandarban, Chittagong, Cox's Bazar, Khagrachari, and Rangamati), most (55%) of the landslides occurred in the Chittagong and Rangamati districts. About 45% of the landslides were small (<100 m$^2$) in size, while the maximum size of the detected landslides was 85202 m$^2$. This dataset will help to understand the characteristics of landslides in CHA and provide useful guidance for policy implementation.

**Dataset:** Dataset is submitted as a supplementary material; link: http://www.mdpi.com/2306-5729/5/1/4/s1.

**Keywords:** landslide; landslide inventory; Chittagong Hilly Areas; attributes of landslides

---

## 1. Introduction

Landslides are the movement of rock, soil, and debris downslope under the influence of gravity [1] and depend on various factors including local geology, topography, climate, and land use/land cover type [2]. Prolonged rainfall and earthquakes are the primary triggers of landslides. Road construction on the slopes, hill cutting, and deforestation are the major anthropogenic activities that create a conducive condition for landslides [3].

Landslide susceptibility mapping is essential to mitigate landslide disasters, and a landslide inventory is the first step toward susceptibility assessment [1]. Since landslides generally occur in existing slide areas, it is vital to know the locations of previously occurred landslides, the size of the landslides, and their-related geomorphological factors [4]. A landslide inventory is a dataset of various information associated with landslides including the absolute and relative location, date, type, size, distribution, casualties, and triggers of landslides [1]. Several methods have been used for landslide inventory mapping including field mapping and visual interpretation of aerial and satellite images [5]. The first step to creating a landslide inventory is to map the exact location of landslides and then construct a dataset of landslides [6]. A good landslide inventory is shareable with the broader scientific community and stakeholders [1].

In Bangladesh, landslides occur mainly in the Chittagong Hilly Areas (CHA) (Figure 1). More than 350 people have died as a result of landslides in CHA in the last three decades [7]. Landslide susceptibility mapping in some parts of this area has already been undertaken [8–10]. In these works, researchers have generated landslide inventories using field mapping, visual interpretation, and automatic recognition of landslides from satellite images [10,11]. Sifa et al. [11] and Comprehensive Disaster Management Plan (CDMP) Phase_II [12] have published landslide inventories in three cities in this area: Cox's Bazar, the Teknaf municipalities, and the Chittagong Metropolitan Area (CMA). Our recent work [5] mapped the landslides of the whole region. This inventory can be used for landslide susceptibility mapping of the entire area, which is very important for land use planning and policymaking. It is of importance to publish and make the data available so that the broader scientific community and policymakers can utilize this dataset for scientific purposes as well as decision making.

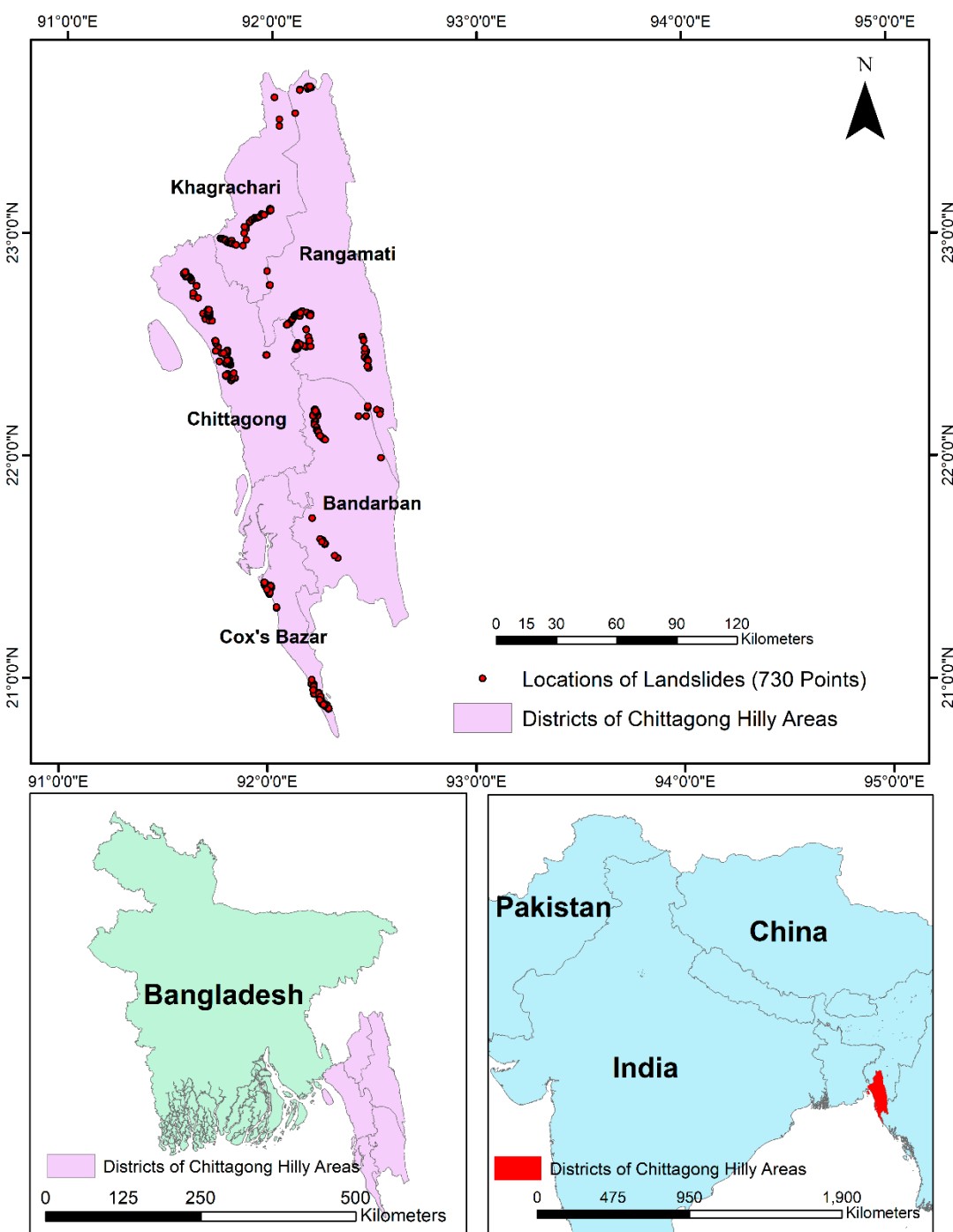

**Figure 1.** Study area and the locations of landslides.

## 2. Data Description

This article describes the landslide inventory dataset of the CHA in Bangladesh (Figure 1). The archived landslides in the inventory took place from January 2001 to March 2017.

*Design of the Dataset*

The dataset was prepared in ArcGIS 10.6.1, and the file types are ESRI shapefiles. The main advantage of using the shapefile format is that the dataset is readily available for working in the ArcGIS platform, thus all necessary statistical and spatial analyses could be conducted with the dataset. Using the ESRI shapefile format, we also did not have to provide the coordinates of the landslides

as these locations are geocoded in the shapefile. We also provided a CSV file of the dataset with the ESRI shapefile for the convenience of users who are not familiar with ArcGIS software. The dataset contains various attributes of landslides (Table 1) such as the type of failure, distribution of landslides, water content and materials, number of deaths, damage caused by the landslides, the total area of the landslide, and the triggers.

**Table 1.** Landslide attributes and data types.

| Type of Attribute | Data Type | Full Explanation | Comment |
| --- | --- | --- | --- |
| ID | Number | Identification Number | |
| District | Text | | The district is the second administrative boundary of Bangladesh. Five districts: Bandarban, Chittagong, Cox's Bazar, Khagrachari, and Rangamati. |
| Location | Text | | Detail address of the landslide location. |
| Fail_Type | Text | Type of failure | Five types (slide, flow, fall, topple and complex) of landslides have been identified based on [13]. The types of 77 landslides were not identified and kept as unrecognized. |
| Date | Text | | Generally, the exact date has been recorded. For Google Earth mapping, the date of the image was recorded. |
| State | Text | Sate of the landslides | Six types of states: active, dormant, inactive, reactivated, stabilized, and suspended. The state of 231 landslides was not determined. |
| Distri_ | Text | Distribution of landslides | Five types of distribution: advancing, diminishing, moving, retrogressive, and widening. The distribution of 286 landslides was not determined. |
| Water_Cont | Text | Water content in the scarp | Two types of water content: wet and dry. The water content of 350 landslides was not determined. |
| Material | Text | The material of the mass moved | Materials include soil, debris, weathered rock and soil, rock. and soil, and a mixture of these materials. For 272 landslides, the material was not determined. |
| Death_ | Text | Number of deaths | |
| Settlemet_ | Text | Number of settlement damaged | |
| Dam_Int1 | Text | Settlement damage intensity | Qualitative judgement (high, medium, and low) of the field investigators. The damage intensity of 271 landslides was not determined. |

**Table 1.** *Cont.*

| Type of Attribute | Data Type | Full Explanation | Comment |
|---|---|---|---|
| **Damae_Int2** | Text | Road damage intensity | Qualitative judgement (high, medium, and low) of the field investigators. The damage intensity of 271 landslides was not determined. |
| **Economic** | Text | Economic loss caused by the landslides | Qualitative judgement (high, medium, and low) of the field investigators. The damage intensity of 271 landslides was not determined. |
| **Area** | Number | Area of landslides | Number of Decimal Places = 0. |
| **Triggers_** | Text | Triggers of landslides | |

## 3. Methodology

There is no standard method to create a landslide inventory dataset. Landslide inventory preparation aims to gather as much data as possible. A dataset includes various attributes, but some attributes may not be available for all landslides [14]. Most landslides are compiled from different sources for different purposes. We adopted three methods in our study: the visual interpretation of Google Earth images, a literature search, and field mapping where each method has advantages and disadvantages. For example, we could detect the landslide location, type, dimension, and date in Google Earth mapping, while for the literature search, we could only obtain data that were recorded in the literature. In contrast, most attributes related to landslide inventories can be gathered in field mapping. The combination of these three methods can help to generate a landslide inventory of events that have occurred recently, in the past, and in inaccessible and remote areas.

In Google Earth, we adopted four criteria to detect landslides: change of vegetation in historical images; the slope and elevation of the area; morphological changes in the images; and the presence of debris [15]. We considered the change in vegetation as the first indicator of landslides in historical images of Google Earth. Landslides remove vegetation, and this can be detected in pre-event and post-event Google Earth images. We checked the slope and elevation of the area in Google Earth using the Add Path tool. Next, we checked the morphological changes and presence of debris. The removal of vegetation can also occur in plain lands, but landslides cannot, which is why we included the slope and elevation in the criteria of detection. When all four criteria were met for a suspected area, we considered it to be a landslide. As above-mentioned, we had previously mapped the landslides from January 2001 to March 2017 and recorded the location, date, type, and dimensions of the landslide. The details of the landslide mapping in Google Earth are given in [5]. We searched the existing literature and newspaper reports before the field mapping. CDMP-II (2012) [12] and Rahman et al. (2016) [16] provided the landslide locations, date, type, causalities, and triggers of landslides for the Chittagong Metropolitan Area (CMP), Cox's Bazar, and Teknaf municipalities. CDMP-II (2012) [12] did not provide the size of landslides, while Rahman et al. (2016) [16] provided the size of the landslides. Newspaper reports (1980–2017), records of the Disaster Management Department of the People's Republic of Bangladesh, and the Roads and Highways Department provided the landslide data of landslides that caused casualties and damage to roads. Based on these reports, we selected Rangamati, Bandarban, Khagrachari, and part of the Chittagong district for field mapping. We adopted participatory field mapping with the help of four field investigators to record the landslide locations and various attributes of landslides (Table 1) including causalities, damages, and economic losses [5,17]. From the literature search, we found the location of landslides, but we did not know the exact locations. Participatory field mapping helped us in this regard since local people knew the exact location of the landslides. The trigger of landslides and financial losses were detected by interviewing local people, government officials, and local political leaders. The damage intensities were identified based on the qualitative judgment of the field investigators (classified into three categories: high, medium, and low). We used

measuring tapes and Global Positioning System (GPS) to measure the area of each landslide. Four well-trained field investigators were hired to collect the type, distribution, state, and water content of landslides through visual investigation using classification schemes outlined in [14,15].

The final dataset was the compilation of the data gathered from field mapping, Google Earth mapping, and the literature search. We also combined the same landslide locations mapped by Google Earth and field mapping.

*Accuracy Assessment*

We assumed that the accuracy of field mapping was better than the other two methods because we visited the field sites and collected GPS locations of the landslides. The accuracy of the field mapping depends on the accuracy of the GPS unit. In this study, we used a Gramin eTrex 20x unit with an accuracy of 3–10 m. The dimensions of the landslides were measured using GPS and measuring tapes. The quality depends on the expertise of the field investigators (in our study, the field investigators were highly trained). The assessments of the damage intensities were qualitative, relying on the capability of field investigators.

We used Google Earth to map landslides and record the data in remote areas, especially in the forests. Due to the remoteness, we did not validate each of the 230 landslides and so carried out the validation in the Bandarban district. We went to the locations of the landslides that we detected in Google Earth for the Bandarban district and verified whether landslides occurred there. We found that the overall accuracy of Google Earth mapping was 88%. This means that 88% of the landslides that we detected in Google Earth were landslides in the Bandarban district. The accuracy of the landslides identified in [12,16] are not known, but we can anticipate a very high accuracy as they used field mapping techniques.

## 4. Inventory Statistics

Most landslides in the dataset occurred in the Chittagong district (208), followed by the Rangamati district (193) (Figure 2). The mean size of the landslides was 1205 m$^2$, with a standard deviation of 5167 m$^2$. The maximum size of the landslide was 85,201 m$^2$, while the minimum size was 11 m$^2$. A total of 45% (Table 2) were small (<100 m$^2$), while 14% of the landslides were within 1000 to 10,000 m$^2$. CDMP II (2012) did not provide landslide areas for 77 landslide locations; thus, the distribution for the size of landslides was based on 653 landslide locations. Slide (285) was the most dominant type of landslides, followed by flow (230), fall (87), complex (34), and topple (17). We failed to recognize the type of 77 landslides, and 62 of them were mapped in Google Earth. Given that the quality of the image was not good enough in 62 landslides, we failed to detect the type. For the remaining 15 landslides, local authorities had removed the debris and reshaped the scarp so that another landslide could not occur. Therefore, we failed to classify these 15 landslides.

**Table 2.** Area of landslides in the CHA, Bangladesh.

| Area of Landslides (m$^2$) | Number of Landslides | Percentage of Landslides |
|---|---|---|
| 0–50 | 185 | 28 |
| 50–100 | 109 | 17 |
| 100–200 | 68 | 10 |
| 200–500 | 102 | 16 |
| 500–1000 | 74 | 11 |
| 1000–10,000 | 91 | 14 |
| 10,000–1,000,000 | 23 | 4 |

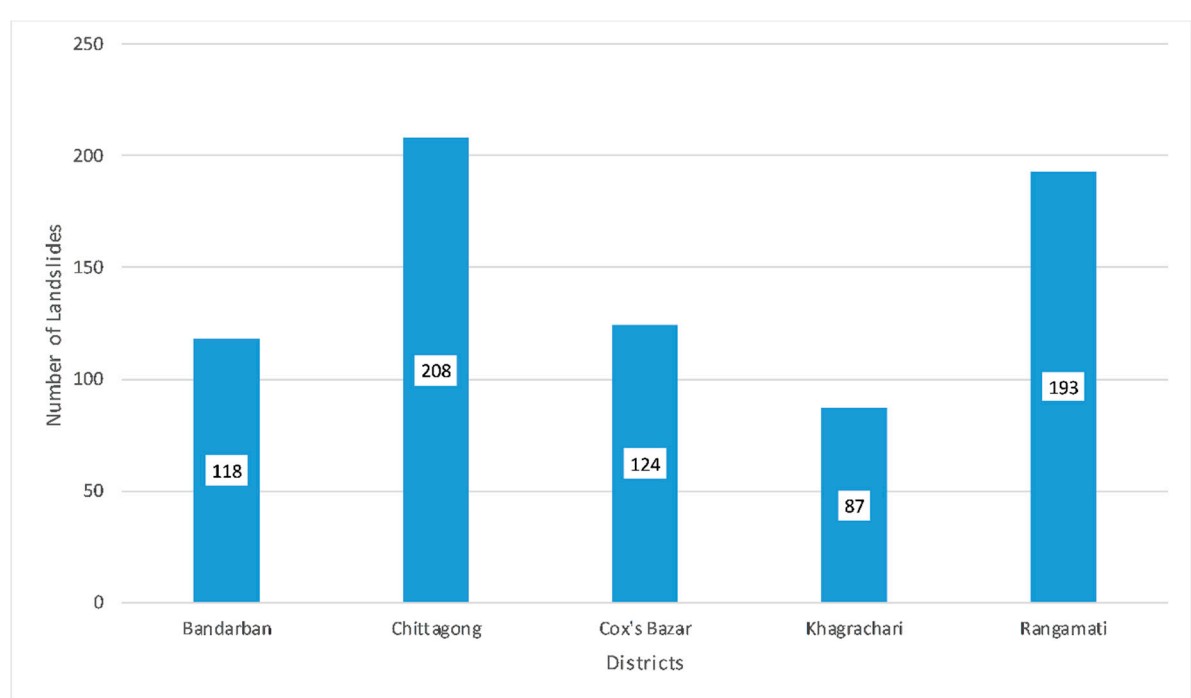

**Figure 2.** Number of landslides in different districts of the Chittagong Hilly Areas (CHA).

The provided dataset is the first landslide inventory that covers the whole region of the CHA, Bangladesh. Since it provides the exact location of landslides, future investigations can select some of the landslide locations to measure slope stability factors and carry out a risk analysis.

**Author Contributions:** Y.W.R. and Y.L. conceived of the idea of this paper. Y.W.R. carried out the field work and Y.L. supervised the whole project. Y.W.R. wrote the manuscript with the support from Y.L. All authors have read and agreed to the published version of the manuscript.

**Funding:** This research was funded by the McClure Scholarship Program of University of Tennessee, Knoxville, USA.

**Conflicts of Interest:** The authors declare no conflicts of interest.

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
