# Peer review of "Landslide Inventory (2001–2017) of Chittagong Hilly Areas, Bangladesh"

_data, 2001_

Round 1

Reviewer 1 Report

Overall this manuscript suffers from poor English and Grammar, but may be of interest to 'Data' readers. I do not believe this goes into enough detail to be of interest to 'Landslide' specialists or Geological Engineers.

Comments by Line #:

9 - ": Landslides are a frequent .."

15 - We often use the term 'triggers' not 'causes' for the initiation of start of a landslide movement.

19 - You state a area with decimal, but your Table states you will not use decimals?

24 - "Landslides are the movement .."

25 - They are not all natural events.

29 - ".. landslide disasters, and a landslide inventory .."

31 - " .. locations of previously occured landslides, the size of .."

32 - "A landslide inventory is a dataset .."

36 - ".. first step of creating a landslide inventory .."

37 - ".. shareable with the broader scientific community .."

43 - Don't start a sentence with a numerical reference. Here use the actual authors then the number reference - "Sifa et al., and CDMP-II [11,12] published landslide inventories in three cities in this area: .."

45 - ".. of the whole region."

FIGURE 1

Needs to be enlarged. The Lat/Longs are unreadable. The North arrows are too small, and you may only need 1. The scales are too small. There are 2 north arrows in the bottom right panel. There are no panel labels. There is no Figure 1 label. There is no caption. I do not see 730 events? What are the dots? What part of the inventory? Perhaps label the actual districts you studied?

52 - ".. mapping, and a literature search."

55 - ".. and the file types are ESRI shapefiles."

58 - ".. such as a CSV file."

60 and below - Why bolded text here? Check Fonts.

64 - Define CDMP II before first use.

65&66 - ".. Slides (285) area the most ... ... flows (230), falls (87), complex (34), and topples (17).

67 - Why did you fail to classify these 77 events? About 10% of your inventory!

FIGURE 2

The y-axis did not appear on my print (odd format maybe). I would shorten x-axis label to just 'Location' or 'District'.

72&73 - bolded text still remains and no need to excessive capitalization.

TABLE 1

200-500 line does not need the '%' symbol. The authors can also probably remove the extra line breaks to make the table smaller.

79 - ".. Earth images, a literature search .. .. . Each method has advantages and .."

82 - ".. to landslide inventories can be gathered in field mapping."

86 - ".. , the slope, elevation of the area, .."

87 - "We mapped landslides from 2001 to March 2017" --What month start in 2001?

88 - "..dimensions of each landslide."

89 - ".. reports before field mapping." Then same issue as #43.

91&92 - Same issue as #43 and #89 ([12] then [11])

92 - ".. [11] presented." I think (presented) is the wrong word here or thought is incomplete?

101 - ".. used measuring tape and GPS .."

102 - ".. trained field investigators .."

103 - ".. classification schemes outlined in [14,15]."

105 - ".. mapping, and a literature search."

108 - ".. the accuracy of field mapping is .."

TABLE 2

Random capitals in the table caption? The right-most column does not need to be centred. It should be left justified like normal text. The authors can also probably remove the extra line breaks to make the table smaller. 2nd last entry in table states no decimal places, but authors used decimals several times prior.

Check your consistency with 'periods' at the end of each reference. Some have them, others do not.

General Comment:

Future Risk is not really addressed, and saying we had experts look at the slopes is not really good enough. They are not even named?

Author Response

Dear Reviewer 1

Thank you for your comments and suggestion. As per your suggestions, we have improved the quality of English and corrected the grammatical errors. We have accepted all the changes that you asked. 

We have addressed some of the issues below which were raised by the reviewer

1.Figure 1

In the study area ma, landslides were presented as points on the map. So, landslides represent a very small area compared to the study area. To make them visible on the map, we used eight symbol size in ArcGIS 10.6.1. It created overlapping points, and thus, it did not look there were 730 landslides on the map. We have reduced the symbol size from 8 to 4. The aim of including the study area map and location of landslides is to give the readers an idea about where landslides previously occurred. We are providing the shapefiles which have the exact location of landslides. Therefore,   we believe it would not be a severe problem if all the locations are not visible on the map. Nevertheless, we have modified the map and included in the text.

2. Figure 2  

We have added a new figure.

3. Future Risk

We have deleted the field " future risk." We have also changed the name of the field causes_ to triggers_

Reviewer 2 Report

An interesting write up of some of the methods used to publish Rabby and Li (2019). I have a few issues which I would like to see addressed before publication. On p. 2, it is stated that "All necessary statistical and spatial analyses can be conducted with the dataset, 57 and it can also be converted into other formats, such as the CSV file." It would be convenient if the authors could do this conversion and make the csv file available so that readers not as familiar with GIS methods and software could have easy access to the data. Also, the next sentence is a little confusing: "Using this format, we do not have to provide coordinates of the landslides as these locations are geocoded in the shapefile". Which format is being referred to here: the shapefile, or the CSV file? The shapefile inherently has the coordinates, and hopefully a csv file would, too. (Indeed, as the data provided are points, not polygons, a flat csv file makes a lot of sense). I would like a little more explanation of the Google Earth analysis, particularly with regard to change over time. Some screen-shots would be illustrative. Some explanation of the participatory field mapping methods (p. 4) would be useful. How many participants were there? How many field investigators were there? On more minor points: p. 1: "More than 350 people have died in CHA in the last three decades" - should read "More than 350 people have died as a result of landslides in the CHA in the last three decades" p. 4: "Gramin Trex 20x": should be "Garmin Etrex 20x".  

Author Response

Thank you for your comments and suggestions. We have accepted all the corrections. We have addressed some of the issues mentioned by reviewer two below

1. CSV Files

We have added the CSV files with the shapefile. Please see the attached zip file. It contains both shape and CSV files. 

2. Explanations of Google Earth Mapping and Participatory Field Mapping

We have added the explanations for the Google Earth mapping in lines 93-98 and participatory field mapping in lines 116-120.